# A Sensitivity-Enhanced Electrolyte-Gated Graphene Field-Effect Transistor Biosensor by Acoustic Tweezers

**DOI:** 10.3390/mi12101238

**Published:** 2021-10-13

**Authors:** Yan Chen, Wenpeng Liu, Hao Zhang, Daihua Zhang, Xiaoliang Guo

**Affiliations:** 1Beijing Engineering Research Center of Industrial Spectrum Imaging, School of Automation and Electrical Engineering, University of Science and Technology Beijing, Beijing 100083, China; yanchen@ustb.edu.cn; 2School of Automation and Electrical Engineering, University of Science and Technology Beijing, Beijing 100083, China; 3College of Precision Instrument and Optoelectronics Engineering, Tianjin University, Tianjin 300072, China; haozhang@tju.edu.cn (H.Z.); dhzhang@tju.edu.cn (D.Z.); 4College of Information Science and Technology, Beijing University of Chemical Technology, Beijing 100029, China

**Keywords:** electrolyte-gated graphene field-effect transistors, acoustic tweezers, solid mounted resonator (SMR)

## Abstract

Low-abundance biomolecule detection is very crucial in many biological and medical applications. In this paper, we present a novel electrolyte-gated graphene field-effect transistor (EGFET) biosensor consisting of acoustic tweezers to increase the sensitivity. The acoustic tweezers are based on a high-frequency bulk acoustic resonator with thousands of MHz, which has excellent ability to concentrate nanoparticles. The operating principle of the acoustic tweezers to concentrate biomolecules is analyzed and verified by experiments. After the actuation of acoustic tweezers for 10 min, the IgG molecules are accumulated onto the graphene. The sensitivities of the EGFET biosensor with accumulation and without accumulation are compared. As a result, the sensitivity of the graphene-based biosensor is remarkably increased using SMR as the biomolecule concentrator. Since the device has advantages such as miniaturized size, low reagent consumption, high sensitivity, and rapid detection, we expect it to be readily applied to many biological and medical applications.

## 1. Introduction

Low-abundance biomolecule detection is very crucial in many biological and medical applications such as clinical diagnostics [1,2], drug discovery [3], and fundamental research [4]. As the conventional optical detection methods require professional skills and complex labeling processes, one of the trends of developing biosensors is exploring miniaturized analytical systems with reduced reagent consumption, high sensitivity, and rapid detection [5]. Various novel methods including surface plasmon resonance (SPR) [6], quartz crystal microbalance (QCM) [7], and electrochemical sensors [8], have been proposed. One of the novel methods is electrical biosensors that employ novel nanomaterials such as silicon nanowires and carbon nanotubes, which has attracted significant attention due to the advantages of its miniaturized size, low reagent consumption, high sensitivity, and rapid detection. Among various types of nanomaterials, graphene, which is a single layer two-dimensional crystal, has emerged as one of the most promising nanoplatforms [5]. Graphene shows extremely high mobility of ∼10^4^ cm^2^ V^−1^ s^−1^ and large carrier capacities of ∼10^12^ cm^−2^, even at room temperature without doping [9,10,11,12,13,14,15]. Meanwhile, as its electrical characteristics are sensitive to the surface conditions [11], the graphene can be used to design novel biosensors with high sensitivity [5,16,17].

Sensitivity is one of the key parameters of modern biosensors [18,19,20]. In this paper, we focus on how to further improve the sensitivity of the graphene-based biosensor. One of the fundamental factors hindering the improvement of the limit of detection is the mass transfer limitations [21,22]. To overcome the mass transfer limitations, various methods, including the electrokinetic-assisted method [23,24], magnetically-assisted method [25], and optically-assisted method [26,27,28], etc., have been proposed. The electrokinetic-assisted technique is limited due to the requirement of inherent charges of targets and the low ionic strength of solutions, thus hindering its applicability to many practical assays. The magnetically- and optically-assisted methods require extra labeling steps or complicated setups, which also limit their throughput.

In this work, we proposed a novel integrated biosensor consisting of an electrolyte-gated graphene field-effect transistor (EGFET) and solid mounted resonator (SMR) as acoustic tweezers. The SMR serves as an active biomolecule concentrator, which was verified in this paper. Thus, the sensitivity of the graphene-based biosensor was greatly enhanced. Meanwhile, the device fabrication is compatible with conventional semiconductor manufacturing processes and system-level integration, which means a low cost and miniaturization. Overall, the proposed method is very promising in biological and medical applications.

## 2. Methods

### 2.1. Mechanism Analysis

Acoustic tweezers are based on the interaction between acoustic waves and target particles within the fluids. One convenient way to generate acoustic waves is to use transducers made of piezoelectric materials. Piezoelectric materials can generate electrical polarization after applying mechanical stress, and the electrical polarization can also lead to mechanical deformation. Due to the piezoelectric effect, the radio frequency (RF) source applied to the SMR device can generate high-frequency longitudinal acoustic waves. When the piezoelectric material is immersed in solution, the high-frequency acoustic waves will trigger the acoustic streaming effects due to acoustic power leakage during the propagation of the acoustic waves into liquid [29,30,31]. High-frequency acoustic waves will contribute to the streaming flow strength due to the great energy attenuation coefficient after their transmission into liquid. Moreover, longitudinal waves can provide more effective energy coupling into the liquid than the shear mode acoustic waves to induce stronger turbulent flow [31]. To be exact, SMR devices have both features, namely high frequency and longitudinal acoustic waves, which can increase the acoustic streaming. Overall, the longitudinal wave and high frequency enable the SMR to be an effective actuator in triggering acoustic streaming and further driving biomolecules such as proteins and DNAs.

The fluid motions excited by the SMR device are visualized in Figure 1a. The longitudinal wave induces an upward fluid flow upon the top inner electrode (TIE) while the surrounding liquids are replenished, thus inducing a localized liquid flow. The biomolecules from the liquid around the device are subsequently transported to the graphene between the TIE and top outer electrode (TOE) by following the flow. Thus, an SMR-induced concentrating effect is generated around the gap between the TIE and TOE, as shown in Figure 1b.

The quantity of the biomolecules absorbed nonspecifically by the EGFET has a positive correlation with the accumulation of the biomolecules. The charges of the absorbed molecules can modulate the drain current of the EGFET; therefore, the Dirac point will drift. The accumulation of the biomolecules around the EGFET can be measured by monitoring the Dirac point. Since the SMR works as a biomolecule concentrator, the accumulation of the biomolecules around the EGFET is much higher than the bulk accumulation, thus lowering the detection limit of EGFET.

### 2.2. Measurement Setup

After graphene patterning, the chip was mounted onto an evaluation board with the TIE and TOE connected to the center pin and outer shell of a coaxial small-A-type (SMA) connector. The board was then connected to a Bias-Tee to split RF and direct-current (DC) signals from each other. The configuration allows the dual modes to operate independently and simultaneously at distinct frequencies (GHz vs. DC) with negligible interference. The two output terminals of the Bias-Tee were connected to a network analyzer (Agilent E5071B, Agilent, Santa Clara, CA, USA) and the source meter (Agilent B1500, Agilent, Santa Clara, CA, USA) separately to characterize the RF and DC responses, respectively. The TOEs were grounded. 

A 100 μL solution reservoir was mounted onto the device, and an Ag/AgCl reference electrode was used as the gate electrode to minimize environmental effects. The gate electrode was connected to the other channel of the Agilent B1500. The schematic diagram of the measurement setup is shown in Figure 1c. 

IgG labeled with green fluorescence (Biosynthesis Biotechnology, Beijing, China) was used for the detection experiments. Various concentrations of IgG solution were obtained by stepwise diluting a certain volume of IgG solution (1 mM) into phosphate buffer saline (PBS). The IgG solution was added into the solution reservoir by pipette. The top-view images of the device were acquired by a microscope (Olympus BX53, Olympus, Tokyo, Japan) with a CCD camera (Olympus DP73, Olympus, Tokyo, Japan).

### 2.3. Fabrication of Device

The devices were fabricated on 100 mm undoped silicon wafers. The alternating aluminum nitride (AlN) and silicon dioxide (SiO_2_) layers were deposited onto the silicon substrate to form the Bragg reflector [32], as shown in Figure 2a. The Bragg reflector transforms the impedance of the substrate to a near-zero or infinite value to avoid wave energy dissipation into the substrate [33]. The thicknesses of each AlN/SiO_2_ pair were 1200/700, 1000/1300, and 1000/650 nm from bottom to top. The bottom electrode (BE) of the SMR was made of 600 nm thick molybdenum (Mo) on top of the Bragg reflector. The film was then patterned into isolated islands. A 1000 nm thick AlN film was then deposited on top of the BE. Orientation of the AlN crystal was along the c-axis. In the final step, the SMR was capped with a pair of gold (Au) top electrodes. The thicknesses of the Au electrodes and the chromium (Cr) adhesive layer were 300 and 50 nm, respectively. The TIE was intentionally shaped like a pentagon to suppress spurious resonance in the device. The electrode area was configured to be 3.0 × 10^4^ μm^2^ so that the SMR has a characteristic impedance of 50 Ω to match the impedance of external circuits. The TOE was separated 15 μm away from the TIE. The area of TOE was several times larger than that of TIE. This geometrical arrangement ensures good electric field confinement within the active area under the TIE electrode. More details have been reported in the literature by us [32].

Graphene field-effect transistors gated by electrolyte were used for detecting the biomolecules. Then, monolayer graphene film was transferred to the SMR. The chemical vapor deposition (CVD) grown monolayer graphene film (VIGON Technologies, Taipei, Taiwan) first went through a thorough RCA cleaning process to remove surface contaminants. Afterward, the device wafer was pretreated with O_2_ plasma to clean the surface and condition it to a hydrophilic interface. The graphene film was then pressed against the wafer with the support of polymethylmethacrylate (PMMA) stamp and held in position for 2 h at room temperature for water evaporation. The fixture was then heated up to 150 °C with a hot plate and kept for 15 min. The heating step promotes van der Waals binding at the graphene–SMR interface, leaving an even and strongly-adhered graphene film after the removal of the PMMA stamp by immersing in acetone for 10 min, as shown in Figure 2b. The graphene film was trimmed using E-beam lithography, followed by O_2_ plasma etching. We used a negative E-beam resist (Allresist, Zwickau, Germany), which introduced much fewer surface contaminants compared to most photoresists. Finally, the biosensor was mounted onto an evaluation board, as shown in Figure 2c.

### 2.4. Characterization of Device

The electrical performance of the device was measured from the RF port of the Bias-Tee, as shown in Figure 3a. The blue and red curves are the magnitude of electrical impedance at a frequency from 1.45 to 1.80 GHz when the device was in the air and immersed in liquid, respectively. The device behaved the same as a bare SMR without graphene in the air [29]. However, we saw a decrease in the Q factor from 275 to 69 when the device was immersed in liquid. The additional energy loss was primarily due to acoustic streaming effects in the liquid. The resonant frequency (1.645 GHz) was chosen to trigger the acoustic streaming effect due to its relatively large acoustic power leakage into the liquid [4]. The transfer characteristics of the EGFET on the SMR device in the liquid were also tested and are shown in Figure 3b. It behaved as a standard bio-polar EGFET. Surface doping of oxygen and moisture from ambient air leads to a p-type channel and shifts the charge neutrality point (CNP) toward positive gate bias of 0.17 V. Hole mobility of the device is derived to be ∼87.4 cm^2^/(V⋅s).

## 3. Results and Discussion

The solution containing green fluorescence-labeled IgG molecules was added into the reservoir. An RF signal of 1.645 GHz at a power of 10 mW was applied to the SMR device to generate acoustic waves. The IgG molecules were driven by acoustic waves to the gap between the TIE and TOE on the graphene. The green fluorescence intensity of the gap was observed to be increased on the gap through a fluorescence microscope, which indicated that the biomolecules were accumulated. After 10-min actuation, the fluorescent intensity of the biomolecules became stable. The final accumulated result is shown in Figure 4b. Compared with the other methods, such as the dielectrophoretic method [34] and ciliated micropillar-based method [35], which requires 30 min and 10 min, respectively, our method possesses an advantage with rapid accumulation.

After accumulation at various concentrations (1 nM, 5 nM, 10 nM, 25 nM, 50 nM, 75 nM, 100 nM, 125 nM, 150 nM, 175 nM, and 200 nM) of the IgG molecule by the SMR device, the Dirac points were calculated by sweeping the V_gs_. Meanwhile, the Dirac points of the EGFETs merged into the same various concentrations of the IgG molecule without accumulation was also measured. Each concentration was measured three times. The Dirac point shifts against the concentration with and without accumulation are shown in Figure 5a. The detection limit of the sensor with accumulation is about 1 nM. Saturation concentrations are 100 nM. The Dirac point shift of the sensor without accumulation shows insignificant change in the concentration’s range (1 nM~200 nM). The Dirac point shift of the sensor without accumulation is 0.37 V at 100,000 nM; this is close to the signal of the sensor without accumulation at 10 nM. We can see that the sensitivity of the device with accumulation is much higher than that without accumulation. The signal of the detection was increased by more than 4 orders. The data are fitted using the linear least-squares fitting method, as shown in Figure 5b. The sensor also has linearity with the R^2^ of 0.96.

## 4. Conclusions

In this paper, we have demonstrated a novel graphene-based transistor biosensor using acoustic tweezers to increase the sensitivity of the biosensor. Since the fabrication steps are compatible with IC processes, which means a small size, low reagent consumption, and low cost, this biosensor can be used as a disposable sensor to ensure repeatability and avoid cross contamination. The operating principle of the device has been described in detail and verified by experiment. We demonstrated that the biomolecules can be concentrated onto the graphene surface efficiently, which greatly increases the sensitivity. Overall, the device has great potential in biological and medical applications.

## Figures and Tables

**Figure 1 micromachines-12-01238-f001:**
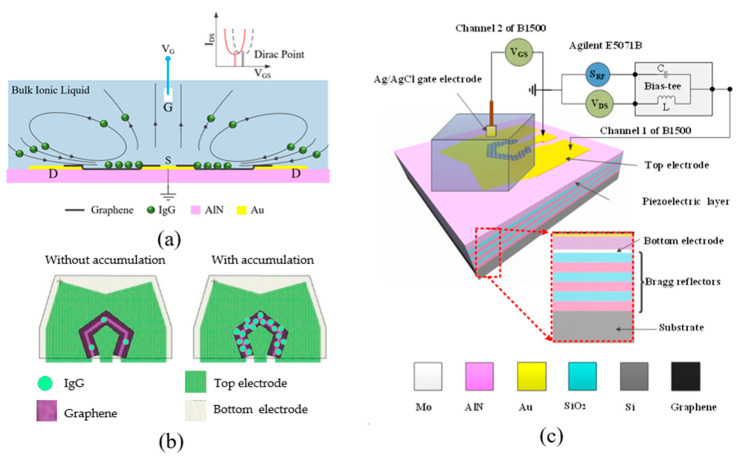
(**a**) The working principle of the EGFET with acoustic tweezers. (**b**) Schematic illustrating the accumulation of IgG molecules using SMR and the detection using EGFET. (**c**) The schematic diagram of the measurement setup.

**Figure 2 micromachines-12-01238-f002:**
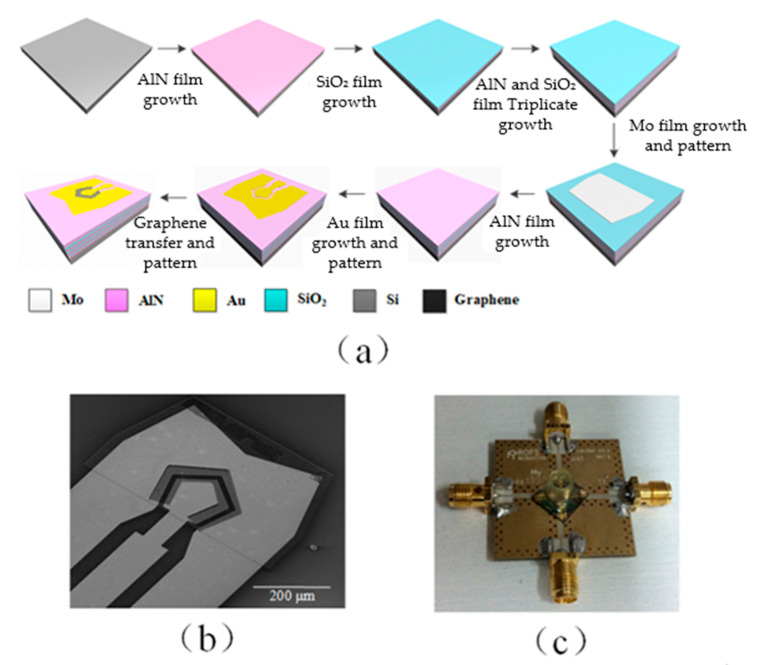
(**a**) The entire process flow SMR fabrication and graphene transfer. (**b**) SEM image of the device. The scale bar is 200 μm. (**c**) Image of a completed device mounted on the evaluation board.

**Figure 3 micromachines-12-01238-f003:**
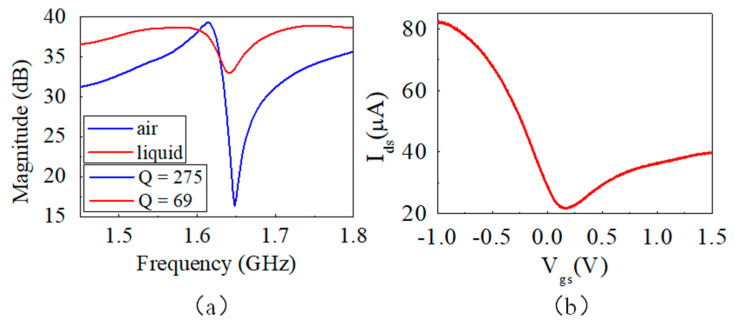
(**a**) Magnitude of electrical impedance at various frequencies and the Q values of the device in the air (blue curve) and in liquid (red curve). (**b**) Transfer characteristics of the graphene FET.

**Figure 4 micromachines-12-01238-f004:**
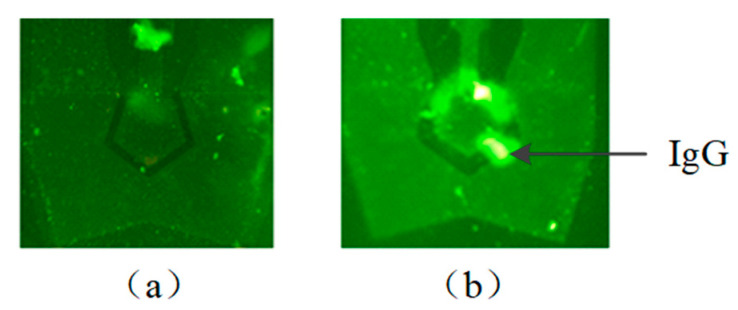
Fluorescence microscope images (**a**) without accumulation (**b**) with accumulation for 10 min.

**Figure 5 micromachines-12-01238-f005:**
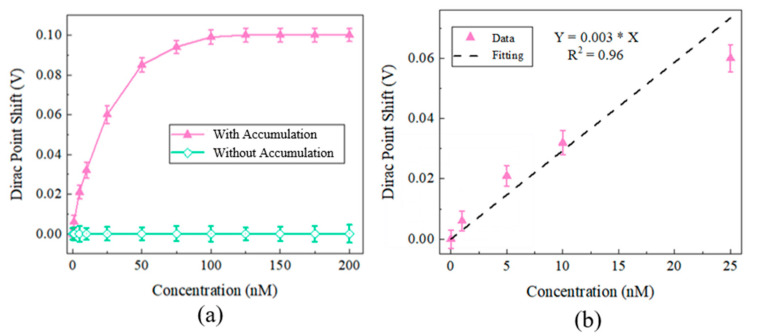
(**a**) Dirac point shifts at various concentrations with and without accumulation. (**b**) The relationship between the detection IgG concentration and the Dirac point shift.

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
