# Peer review of "A Sensitivity-Enhanced Electrolyte-Gated Graphene Field-Effect Transistor Biosensor by Acoustic Tweezers"

_micromachines, 2021, doi:10.3390/mi12101238_

Round 1
Reviewer 1 Report
The paper
A Sensitivity-Enhanced Electrolyte-Gated Graphene Field-Ef-2 fect Transistors Biosensor by an Acoustic Tweezers 3
by
Yan Chen, Wenpeng Liu, Hao Zhang , Daihua Zhang,and Xiaoliang Guo
reports on h high sensitivity biosensor based on a field effectector on graphene. On their sensor, the sensitivity increase is achived via an electrical field induced concentration increase of bio molecules in the detection area.
In principle, I like this work, since any progress in biodetection. especial on an electronic basis is beneficial for all healts applications. Before the paper can be published, however, severals flaws and errors need to be fixed.
1) Several Language problems / typos can be found in this work, e,g,
singular - plural mistakes: tweezer - tweezers
104 cm^2 -> 10^4 cm^2
1012 cm^-2 -> 10^12 cm^-2
a great sensitivity -> very large sensitivity
Daric points -> Dirac points (2x)
3.3. Biomolecur concentrating -> Biomolecular ?
Please cross check your paper carefully.
2) Some detailed remarks on the scientific content:
Fig. 2:
What is this Bragg-reflector good for? At least one sentence for non-experts would be nice.
Figure 5:
…. 10 min….
“10 min” is not very fast. How does this compare with other devices of similar purpose ?
The largest problem (due to language problems, I assume), however, is the title of the paper and the wording in the following. The title: “ A Sensitivity-Enhanced Electrolyte-Gated Graphene Field-Effect Transistors Biosensor by an Acoustic Tweezers " is completely unappropriate in my opinion, since there are no acoustic effects in this work. All described effects are due to microwaves and electric fields. I suggest to change the title into-> ----“RF - optical tweezer” … or a similar formulation.
Author Response
Reviewer 1
Point 1:
Several Language problems / typos can be found in this work, e,g,
singular - plural mistakes: tweezer - tweezers
104 cm^2 -> 10^4 cm^2
1012 cm^-2 -> 10^12 cm^-2
a great sensitivity -> very large sensitivity
Dirac points -> Dirac points (2x)
3.3. Biomolecur concentrating -> Biomolecular ?
Please cross check your paper carefully.
Response:
Thanks for the suggestions. Several language problems was modified and we inserted some recent works.
Point 2:
Some detailed remarks on the scientific content:
Fig. 2:
What is this Bragg-reflector good for? At least one sentence for non-experts would be nice.
Figure 5:
…. 10 min….
“10 min” is not very fast. How does this compare with other devices of similar purpose ?
Response:
Thanks for the positive comments. The Bragg reflector transforms the impedance of the substrate to a near-zero or infinite value to avoid wave energy dissipation into the substrate [33]. This sentence explains and the corresponding reference was inserted in the recent manuscript in Line 122-123.
Compared with the methods reported below, dielectrophoretic (DEP) separation 30 min [34], Ciliated micropillars isolation 10 min [35], it has obvious time advantages. This sentence explains and the corresponding reference was inserted in the recent manuscript in Line 183-184.
Point 3:
The largest problem (due to language problems, I assume), however, is the title of the paper and the wording in the following. The title: “A Sensitivity-Enhanced Electrolyte-Gated Graphene Field-Effect Transistors Biosensor by an Acoustic Tweezers " is completely unappropriate in my opinion, since there are no acoustic effects in this work. All described effects are due to microwaves and electric fields. I suggest to change the title into-> ----“RF - optical tweezer” … or a similar formulation.
Response:
Thanks for the careful review.
Acoustic tweezers are based on the interaction between acoustic waves and tar-gets within the fluids. One convenient way to generate acoustic waves is to use trans-ducers made of piezoelectric materials. Piezoelectric materials can generate electrical polarization under applied mechanical stress, and the electrical polarization can lead tomechanical deformation. Due to the piezoelectric effect. These sentences explain are inserted in the recent manuscript in Line 63-67.
Reviewer 2 Report
In this work, the authors used a previously developed device, which is combination of a thin-film piezoelectric resonator and a graphene FET and investigated the effect of the micro-vortices and acoustic streaming generated by the resonator on sensor sensitivity. Biomolecule accumulation was observed by fluorescence microcopy and electrically detected by means of the Dirac point shift.
The paper is well organized and clear, and is suitable for publication, after a revision of the English usage and grammar. Several corrections are necessary, for example:
Line 7: "Electeial" to "Electrical".
Line 22: "regent" to "reagent"
Figure 2: "baistee" to "bias-tee", "bottle" to "bottom"
Line 136: "visualize" to "visualized".
Lines 162, 183, 188, 205: "Daric" and "Diract" to "Dirac".
Line 169: "Biomolecur" to "biomolecule".
Line 185: “time” to “times”.
Figure 7, legend: "Date" to "data".
Figure 6, legend: “concentraing” to “concentration”.
Additional corrections:
1) Lines 40 and 41: The exponents "12" and "4" should appear in superscript. Also check line 73: "104".
2) Reference to Figure 1 is missing.
Author Response
Reviewer 2
Point 1:
Line 7: "Electeial" to "Electrical".
Line 22: "regent" to "reagent"
Figure 2: "baistee" to "bias-tee", "bottle" to "bottom"
Line 136: "visualize" to "visualized".
Lines 162, 183, 188, 205: "Daric" and "Diract" to "Dirac".
Line 169: "Biomolecur" to "biomolecule".
Line 185: “time” to “times”.
Figure 7, legend: "Date" to "data".
Figure 6, legend: “concentraing” to “concentration”.
Response:
Thanks for the thorough review and comments. Several problems were modified and we inserted some recent works.
Point 2:
Additional corrections:
1) Lines 40 and 41: The exponents "12" and "4" should appear in superscript. Also check line 73: "104".
2) Reference to Figure 1 is missing.
Response:
Thank you for your advice. Several problems were modified and we inserted some recent works.
Reviewer 3 Report
The authors have presented an electrolyte-gated GFET biosensor using acoustic tweezers. While the research is interesting, the language/writing style needs to be significantly improved before this article can be considered for publication. Apart from that, I have some major concerns regarding the article.
(1) The working principle of the device is not explained. The authors showed the description of the device and the fabrication process, but the purpose and motivation behind this particular device design are not explained.
(2) The basic electrical and material characterization are missing. The quality of graphene will play a significant role, so that information is necessary.
(3) In the measurements section, the authors used the word "concentrating", which is not very clear. Sections 3.3 and 3.4 are very hard to follow due to poor write-up, which needs to be improved. I guess the authors are trying to imply "accumulation", but that needs to be cleared up. Also, they wrote Dirac point as Daric point in few places.
(4) The article lacks critical data such as how well the device recovers after sensing, repeatability, long time stability, etc.
Author Response
Reviewer 3
The authors have presented an electrolyte-gated GFET biosensor using acoustic tweezers. While the research is interesting, the language/writing style needs to be significantly improved before this article can be considered for publication. A part from that, I have some major concerns regarding the article.
Point 1:
The working principle of the device is not explained. The authors showed the description of the device and the fabrication process, but the purpose and motivation behind this particular device design are not explained.
Response:
Thanks for the positive comments. We adjusted the structure of the manuscript, as shown below:
2.Methods
2.1. Mechanism analysis
2.2. Measurement setup
2.3. Fabrication of device
2.4. Characterization of device
The part of mechanism analysis is explained before the part of the fabrication process, the research motivation and purpose of the research was explained.
These adjusted parts were shown in Line 61-173.
Point 2:
The basic electrical and material characterization are missing. The quality of graphene will play a significant role, so that information is necessary.
Response:
Thank you for your advice. Surface doping of oxygen and moisture from ambient air leads to a p-type channel and shifts the charge neutrality point (CNP) toward positive gate bias The Dirac point of the GFET on the SMR in liquid is 0.17 V, Hole mobility of the device is derived to be ∼87.4 cm2/(V s).
The basic electrical and material characterization was inserted in some recent works, as shown in Line166-168.
Point 3:
In the measurements section, the authors used the word "concentrating", which is not very clear. Sections 3.3 and 3.4 are very hard to follow due to poor write-up, which needs to be improved. I guess the authors are trying to imply "accumulation", but that needs to be cleared up. Also, they wrote Dirac point as Daric point in few places.
Response:
Thanks for the careful review. There is biomolecule accumulation, we adjusted the structure of the manuscript, combine sections 3.3 and 3.4.
The solution containing green fluorescence-labled IgG molecules was added into the reservoir. An RF signal of 1.645 GHz at a power of 10 mW was applied to the SMR device to generate acoustic waves. The IgG molecules were driven by acoustic waves to the gap between the TIE and TOE on the graphene. The green fluorescence intensity of the gap was observed to be increased on the gap through a fluorescence microscope, which indicated that the biomolecules were accumulated. After 10-minute actuation, the fluorescent intensity of the biomolecule became stable. The final accumulated re-sult is shown in Figure 4(b). Compared with the methods reported below, dielectro-phoretic (DEP) separation 30 min [34], ciliated micropillars isolation 10 min [35], it has obvious time advantages.
After concentration various concentrations (1 nM, 5 nM, 10 nM, 25 nM, 50 nM, 75 nM, 100 nM, 125 nM, 150 nM, 175 nM and 200 nM) of the IgG molecule by the SMR de-vice, the Dirac points were calculated by sweeping the Vgs. Meanwhile, the Dirac points of the EGFETs merged into the same various concentrations of the IgG molecule without concentrating was also measured. Each concentration was measured three times. The Dirac point shifts against the concentration with and without concentration are shown in Figure 5(a). The detection limit of the concentration sensor is about 1 nM. Saturation concentrations are 100 nM. The Dirac point shift without concentration sensor shows insignificant change in concentrations range (1 nM ~ 200 nM). The Dirac point shift of the without concentration sensor is 0.37 V at 100000 nM, it is close to the signal with concentration sensor at 10 nM. We can see that the sensitivity of the device with concentration is much higher than that without concentration. The signal of the detection was increased by more than 4 orders. The data were fitted using the linear least-squares fitting method is shown in Figure 5(b). The sensor also has linearity with the R2 of 0.96.
These sentences explain are inserted in the recent manuscript in Line 174-205.
Several wrote Daric point was modified to Dirac point.
Point 4:
The article lacks critical data such as how well the device recovers after sensing, repeatability, long time stability, etc.
Response:
Thank you for your advice. Since the device can be fabricated with low cost and high throughput, it can be used as a disposable sensor without regeneration to ensure the repeatability and long time stability. Meanwhile, this disposable sensors can also avoid the cross contamination.
To clarify this, we explained this in the Conclusion section in the main text.
Round 2
Reviewer 1 Report
The revised version of the paper has clearly improved, however, two minor things must be corrected before publication:
Chapter 2.1, first paragraph: Spelling mistake: metarial -> material
Fig. 5 consists of 4 sub-figures and the labeling (a) (b) of the sub-figures is inconsistent with the figure captions.
The formulation ".....at various concentrations with and without concentration...." is misleading.
Author Response
Reviewer 1
The revised version of the paper has clearly improved, however, two minor things must be corrected before publication.
Point 1:
Chapter 2.1, first paragraph: Spelling mistake: metarial -> material
Response:
Thanks for the suggestions. The problem was modified and we inserted some recent works.
Point 2:
Fig. 5 consists of 4 sub-figures and the labeling (a) (b) of the sub-figures is inconsistent with the figure captions.
Response:
Thanks for the careful review. We have updated the Figure as follow:
Figure 5. (a) Dirac point shifts at various concentrations with and without accumulation. (b) The relationship between the detection IgG concentration and the Diract point shift.
Point 3:
The formulation ".....at various concentrations with and without concentration...." is misleading.
Response:
Thanks for the positive comments. To avoid the misleading, we have updated this sentence as: Dirac point shifts at various concentrations with and without accumulation. This sentence was inserted in the recent manuscript in Line 209.
Reviewer 3 Report
I think the reviewers have addressed the issues quite well in the revision, therefore I recommend the publication of this manuscript after minor spell/grammar checks.
Author Response
Reviewer 2
I think the reviewers have addressed the issues quite well in the revision, therefore I recommend the publication of this manuscript after minor spell/grammar checks.
Response:
Thanks for the thorough review and comments. We have revised the paper throughout to reduce the spelling and grammar errors as possible as we can.